# Beyond Vector Spaces: Compact Data Representation as Differentiable Weighted Graphs

**Denis Mazur**[*]
Yandex
denismazur@yandex-team.ru

**Vage Egiazarian**[*]
Skoltech
Vage.egiazarian@skoltech.ru

**Stanislav Morozov**[*]
Yandex
Lomonosov Moscow State University
stanis-morozov@yandex.ru

**Artem Babenko**
Yandex
National Research University
Higher School of Economics
artem.babenko@phystech.edu

## Abstract

Learning useful representations is a key ingredient to the success of modern machine learning. Currently, representation learning mostly relies on embedding data into Euclidean space. However, recent work has shown that data in some domains is better modeled by non-euclidean metric spaces, and inappropriate geometry can result in inferior performance. In this paper, we aim to eliminate the inductive bias imposed by the embedding space geometry. Namely, we propose to map data into more general non-vector metric spaces: a weighted graph with a shortest path distance. By design, such graphs can model arbitrary geometry with a proper configuration of edges and weights. Our main contribution is PRODIGE: a method that learns a weighted graph representation of data end-to-end by gradient descent. Greater generality and fewer model assumptions make PRODIGE more powerful than existing embedding-based approaches. We confirm the superiority of our method via extensive experiments on a wide range of tasks, including classification, compression, and collaborative filtering.

## 1 Introduction

Nowadays, representation learning is a major component of most data analysis systems; this component aims to capture the essential information in the data and represent it in a form that is useful for the task at hand. Typical examples include word embeddings [1, 2, 3], image embeddings [4, 5], user/item representations in recommender systems [6] and others. To be useful in practice, representation learning methods should meet two main requirements. Firstly, they should be effective, i.e., they should not lose the information needed to achieve high performance in the specific task. Secondly, the constructed representations should be efficient, e.g., have small dimensionality, sparsity, or other structural constraints, imposed by the particular machine learning pipeline. In this paper, we focus on compact data representations, i.e., we aim to achieve the highest performance with the smallest memory consumption.

Most existing methods represent data items as points in some vector space, with the Euclidean space $\mathbb{R}^n$ being a "default" choice. However, several recent works [7, 8, 9] have demonstrated that the quality of representation is heavily influenced by the geometry of the embedding space. In particular, different space curvature can be more appropriate for different types of data [8]. While some prior

---

[*]Equal contribution

works aim to learn the geometrical properties of the embedding space from data[8], all of them assume a vectorial representation of the data, which may be an unnecessary inductive bias.

In contrast, we investigate a more general paradigm by proposing to represent finite datasets as weighted graphs equipped with a shortest path distance. It can be shown that such graphs can represent any geometry for a finite dataset[10] and can be naturally treated as finite metric spaces. Specifically, we introduce **Probabilistic Differentiable Graph Embeddings (PRODIGE)**, a method that learns a weighted graph from data, minimizing the problem-specific objective function via gradient descent. Unlike existing methods, PRODIGE does not learn vectorial embeddings of the data points, instead information from the data is effectively stored in the graph structure. Via extensive experiments on several different tasks, we confirm that, in terms of memory consumption, PRODIGE is more efficient than its vectorial counterparts.

To sum up, the contributions of this paper are as follows:

1. We propose a new paradigm for constructing representations of finite datasets. Instead of constructing pointwise vector representations, our proposed PRODIGE method represents data as a weighted graph equipped with a shortest path distance.
2. Applied to several tasks, PRODIGE is shown to outperform vectorial embeddings when given the same memory budget; this confirms our claim that PRODIGE can capture information from the data more effectively.
3. The PyTorch source code of PRODIGE is available online[1].

The rest of the paper is organized as follows. We discuss the relevant prior work in Section 2 and describe the general design of PRODIGE in Section 3. In Section 4 we consider several practical tasks and demonstrate how they can benefit from the usage of PRODIGE as a drop-in replacement for existing vectorial representation methods. Section 5 concludes the paper.

## 2   Related work

In this section, we briefly review relevant prior work and describe how the proposed approach relates to the existing machine learning concepts.

**Embeddings.** Vectorial representations of data have been used in machine learning systems for decades. The first representations were mostly hand-engineered and did not involve any learning, e.g. SIFT[11] and GIST[12] representations for images, and n-gram frequencies for texts. The recent success of machine learning is largely due to the transition from the handcrafted to learnable data representations in domains, such as NLP[1, 2, 3], vision[4], speech[13]. Most applications now use learnable representations, which are typically vectorial embeddings in Euclidean space.

**Embedding space geometry.** It has recently been shown that Euclidean space is a suboptimal model for several data domains [7, 8, 9]. In particular, spaces with a hyperbolic geometry are more appropriate to represent data with a latent hierarchical structure, and several papers investigate hyperbolic embeddings for different applications [14, 15, 16]. The current consensus appears to be that different geometries are appropriate for solving different problems and there have been attempts to learn these geometrical properties (e.g. curvature) from data [8]. Instead, our method aims to represent data as a weighted graph with a shortest path distance; this by design can express an arbitrary finite metric space.

**Connections to graph embeddings.** Representing the vertices of a given graph as vectors is a long-standing problem in machine learning and complex networks communities. Modern approaches to this problem rely on graph theory and/or graph neural networks [17, 18], which are both areas of active research. In some sense, we aim to solve the inverse problem; given data entities, our goal is to learn a graph approximating the distances that satisfy the requirements of the task at hand.

**Existing works on graph learning.** We are not aware of prior work that proposes a general method to represent data as a weighted graph for any differentiable objective function. Probably, the closest work to ours is [19]; they solve the problem of distance-preserving compression via a dual optimization problem. Their proposed approach lacks end-to-end learning and does not generalize to arbitrary loss functions. There are also several approaches that learn graphs from data for specific problems.

Some studies[20, 21] learn specialized graphs that perform clustering or semi-supervised learning. Others[22, 23] focus specifically on learning probabilistic graphical model structure. Most of the proposed approaches are highly problem-specific and do not scale well to large graphs.

## 3 Method

In this section we describe the general design of PRODIGE and its training procedure.

### 3.1 Learnable Graph Metric Spaces

Consider an undirected weighted graph $G(V, E, w)$, where $V = \{v_0, v_1, \ldots, v_n\}$ is a set of vertices, corresponding to data items, and $E = \{e_0, e_1, \ldots, e_m\}$ is a set of edges, $e_i = e(v_i^{source}, v_i^{target})$. We use $w_\theta(e_i)$ to denote non-negative edge weights, which are learnable parameters of our model.

Our model includes another set of learnable parameters that correspond to the probabilities of edges in the graph. Specifically, for each edge $e_i$, we define a Bernoulli random variable $b_i \sim p_\theta(b_i)$ that indicates whether an edge is present in the graph $G$. For simplicity, we assume that all random variables $b_i$ in our model are independent. In this case the joint probability of all edges in the graph can be written as $p(G) = \prod_{i=0}^{m} p_\theta(b_i)$.

The expected distance between any two vertices $v_i$ and $v_j$ can now be expressed as a sum of edge weights along the shortest path:

$$\underset{G \sim p(G)}{E} d_G(v_i, v_j) = \underset{G \sim p(G)}{E} \quad \min_{\pi \in \Pi_G(v_i, v_j)} \sum_{e_i \in \pi} w_\theta(e_i) \tag{1}$$

Here $\Pi_G(v_i, v_j)$ denotes the set of all paths from $v_i$ to $v_j$ over the edges of $G$, or more formally, $\Pi_G(v_i, v_j) = \{\pi : (e(v_i, v_{...}), \ldots, e(v_{...}, v_j))\}$. For a given $G$, the shortest path can be found exactly using Dijkstra's algorithm. Generally speaking, a shortest path is not guaranteed to exist, e.g., if $G$ is disconnected; in this case we define $d_G(v_i, v_j)$ to be equal to a sufficiently large constant.

The parameters of our model must satisfy two constraints: $w_\theta(e_i) \geq 0$ for weights and $0 \leq p_\theta(e) \leq 1$ for probabilities. We avoid constrained optimization by defining $w_\theta(e_i)$ as $softplus(\theta_{w,i})$[24] and $p_\theta(b_i)$ as $\sigma(\theta_{b,i})$.

This model can be trained by minimizing an arbitrary differentiable objective function with respect to parameters $\theta = \{\theta_w, \theta_b\}$ directly by gradient descent. We explore several problem-specific objective functions in Section 4.

### 3.2 Sparsity

In order to learn a compact representation, we encourage the algorithm to learn sparse graphs by imposing additional regularization on $p(G)$. Namely, we employ a recent sparsification technique proposed in [25, 26]. Denoting the task-specific loss function as $L(G, \theta)$, the regularized training objective can be written as:

$$\mathcal{R}(\theta) = \underset{G \sim p(G)}{E} [L(G, \theta)] + \lambda \cdot \frac{1}{|E|} \sum_{i=1}^{|E|} p_\theta(b_i = 1) \tag{2}$$

Intuitively, the term $\frac{1}{|E|} \sum_{i=1}^{|E|} p_\theta(b_i = 1)$ penalizes the number of edges being used, on average. It effectively encourages sparsity by forcing the edge probabilities to decay over time. For instance, if a certain edge has no effect on the main objective $L(G, \theta)$ (e.g., the edge never occurs on any shortest path), the optimal probability of that edge being present is exactly zero. The regularization coefficient $\lambda$ affects the "zeal" of edge pruning, with larger values of $\lambda$ corresponding to greater sparsity of the learned graph.

We minimize this regularized objective function (2) by stochastic gradient descent. The gradients $\nabla_{\theta_w} \mathcal{R}$ are straightforward to compute using existing autograd packages, such as TensorFlow or PyTorch. The gradients $\nabla_{\theta_b} \mathcal{R}$ are, however, more tricky and require the log-derivative trick[27]:

$$\nabla_{\theta_b} \mathcal{R} = \underset{G \sim p(G)}{E} \left[ L(G, \theta) \cdot \nabla_{\theta_b} \log p(G) \right] + \lambda \cdot \frac{1}{|E|} \sum_{i=1}^{|E|} \nabla_{\theta_b} p_\theta(b_i = 1) \quad (3)$$

In practice, we can use a Monte-Carlo estimate of gradient (3). However, the variance of this estimate can be too large to be used in practical optimization. To reduce the variance, we use the fact that the optimal path usually contains only a tiny subset of all possible edges. More formally, if the objective function only depends on a subgraph $\hat{G} \in G : L(\hat{G}, \theta) = L(G, \theta)$, then we can integrate out all edges from $G \setminus \hat{G}$:

$$\underset{G \sim p(G)}{E} L(G, \theta) \cdot \nabla_{\theta_b} \log p(G) = \underset{\hat{G} \sim p(\hat{G})}{E} L(\hat{G}, \theta) \cdot \nabla_{\theta_b} \log p(\hat{G}) \quad (4)$$

The expression (4) allows for an efficient training procedure that only samples edges that are required by Dijkstra's algorithm. More specifically, on every iteration the path-finding algorithm selects a vertex $v_i$ and explores its neighbors by sampling $b_i \sim p_\theta(b_i)$ corresponding to the edges that are potentially connected to $v_i$. In this case, the size of $\hat{G}$ is proportional to the number of iterations of Dijkstra's algorithm, which is typically lower than the number of vertices in the original graph $G$.

Finally, once the training procedure converges, most edges in the obtained graph are nearly deterministic: $p_\theta(b_i = 1) < \varepsilon \vee p_\theta(b_i = 1) > 1 - \varepsilon$. We make this graph exactly deterministic by keeping only the edges with probability greater or equal to $0.5$.

### 3.3 Scalability

As the total number of edges $|E|$ in a complete graph grows quadratically with the number of vertices $|V|$, memory consumption during PRODIGE training on large datasets can be infeasible. To reduce memory requirements, we explicitly restrict a subset of possible edges and learn probabilities only for edges from this subset. The subset of possible edges is constructed by a simple heuristic described below.

First, we add an edge from each data item to $k$ most similar items in terms of problem-specific similarity in the original data space. Second, we also add $r$ random edges between uniformly chosen source and destination vertices. Overall, the size of the constructed subset scales linearly with the number of vertices $|V|$, which makes training feasible for large-scale datasets. In our experiments, we observe that increasing the number of possible edges improves the model performance until some saturation point, typically with $32-100$ edges per vertex.

Finally, we observed that the choice of an optimization algorithm is crucial for the convergence speed of our method. In particular, we use SparseAdam[28] as it significantly outperformed other sparse SGD alternatives in our experiments.

## 4 Applications

In this section, we experimentally evaluate the proposed PRODIGE model on several practical tasks and compare it to task-specific baselines.

### 4.1 Distance-preserving compression

Distance-preserving compression learns a compact representation of high-dimensional data that preserves the pairwise distances from the original high-dimensional space. The learned representations can then be used in practical applications, e.g., data visualization.

**Objective.** We optimize the squared error between pairwise distances in the original and compressed spaces. Since PRODIGE graphs are stochastic, we minimize this objective in expectation over edge probabilities:

$$\underset{G\sim p(G)}{E} L(G,\theta) = \underset{G\sim p(G)}{E} \frac{1}{N^2} \sum_{i=0}^{N} \sum_{j=0}^{N} \left( \|x_i - x_j\|_2 - d_G(v_i, v_j) \right)^2 \qquad (5)$$

In the formula above, $x_i, x_j \in X$ are objects in the original high-dimensional Euclidean space and $v_i, v_j \in V$ are the corresponding graph vertices. Note that the objective (5) resembles the well-known Multidimensional Scaling algorithm[29].

However, the objective (5) does not account for the graph size in the learned model. This can likely lead to a trivial solution since the PRODIGE graph can simply memorize $w_\theta(e(v_i, v_j)) = \|x_i - x_j\|_2$. Therefore, we use the sparsification technique described earlier in Section 3.2. We also employ the initialization heuristic from Section 3.3 to speed up training. Namely, we start with 64 edges per vertex, half of which are links to the nearest neighbors and the other half are random edges.

**Experimental setup.** We experiment with three publicly available datasets:

- **MNIST10k:** $N=10^4$ images from the test set of the MNIST dataset, represented as 784-dimensional vectors;
- **GLOVE10k:** top-$N=10^4$ most frequent words, represented as 300-dimensional pre-trained[2] GloVe[2] vectors;
- **CelebA10K:** 128-dimensional embeddings of $N=10^4$ random face photographs from the CelebA dataset, produced by deep CNN[3].

In these experiments, we aim to preserve the Euclidean distances (5) for all datasets. Note, however, that any distance function can be used in PRODIGE.

| Method | #parameters per instance | #parameters total | MNIST10k | GLOVE10k | CelebA10k |
|---|---|---|---|---|---|
| $\leq$ **4 parameters per instance** | | | | | |
| PRODIGE | $3.92 \pm 0.02$ | 39.2k | **0.00938** | **0.03289** | **0.00374** |
| MDS | 4 | 40k | 0.05414 | 0.13142 | 0.01678 |
| Poincare MDS | 4 | 40k | 0.04683 | 0.11386 | 0.01649 |
| PCA | 4 | 40k | 0.30418 | 0.84912 | 0.09078 |
| $\leq$ **8 parameters per instance** | | | | | |
| PRODIGE | $7.65 \pm 0.14$ | 76.5k | **0.00886** | **0.02856** | **0.00367** |
| MDS | 8 | 80k | 0.01857 | 0.05584 | 0.00621 |
| Poincare MDS | 8 | 80k | 0.01503 | 0.04839 | 0.00619 |
| PCA | 8 | 80k | 0.16237 | 0.62424 | 0.05298 |

Table 1: Comparison of distance-preserving compression methods for two memory budgets. We report the mean squared error between pairwise distances in the original space and for learned representations.

We compare our method with three baselines, which construct vectorial embeddings:

- **Multidimensional Scaling (MDS)[29]** is a well-known visualization method that minimizes the similar distance-preserving objective (5), but maps datapoints into Euclidean space of a small dimensionality.
- **Poincare MDS** is a version of MDS that maps data into Poincare Ball. This method approximates the original distance with a hyperbolic distance between learned vector embeddings: $d_h(x_i, x_j) = arccosh\left(1 + 2\frac{\|x_i - x_j\|_2^2}{(1-\|x_i\|_2^2)\cdot(1-\|x_j\|_2^2)}\right)$

- **PCA.** Principal Component Analysis is the most popular techinque for data compression. We include this method for sanity check.

For all the methods, we compare the performance given the same memory budget. Namely, we investigate two operating points, corresponding to four and eight 32-bit numbers per data item. For embedding-based baselines, this corresponds to 4-dimensional and 8-dimensional embeddings, respectively. As for PRODIGE, it requires a total of $N+2|E|$ parameters where $N=|V|$ is a number of objects and $|E|$ is the number of edges with $p_\theta(b_i) \geq 0.5$. The learned graph is represented as a sequence of edges ordered by the source vertex, and each edge is stored as a tuple of target vertex (int32) and weight (float32). We tune the regularization coefficient $\lambda$ to achieve the overall memory consumption close to the considered operating points. The distance approximation errors for all methods are reported in Table 1, which illustrates that PRODIGE outperforms the embedding-based counterparts by a large margin. These results confirm that the proposed graph representations are more efficient in capturing the underlying data geometry compared to vectorial embeddings. We also verify the robustness of our training procedure by running several experiments with different random initializations and different initial numbers of neighbors. Figure 2 shows the learning curves of PRODIGE under various conditions for **GLOVE10K** dataset and four numbers/vertex budget. While these results exhibit some variability due to optimization stochasticity, the overall training procedure is stable and robust.

**Qualitative results.** To illustrate the graphs obtained by learning the PRODIGE model, we train it on a toy dataset containing 100 randomly sampled MNIST images of five classes. We start the training from a full graph, which contains 4950 edges, and increase $\lambda$ till the moment when only $5\%$ of edges are preserved. The tSNE[30] plot of the obtained graph, based on $d_G(\cdot, \cdot)$ distances, is shown on Figure 1.

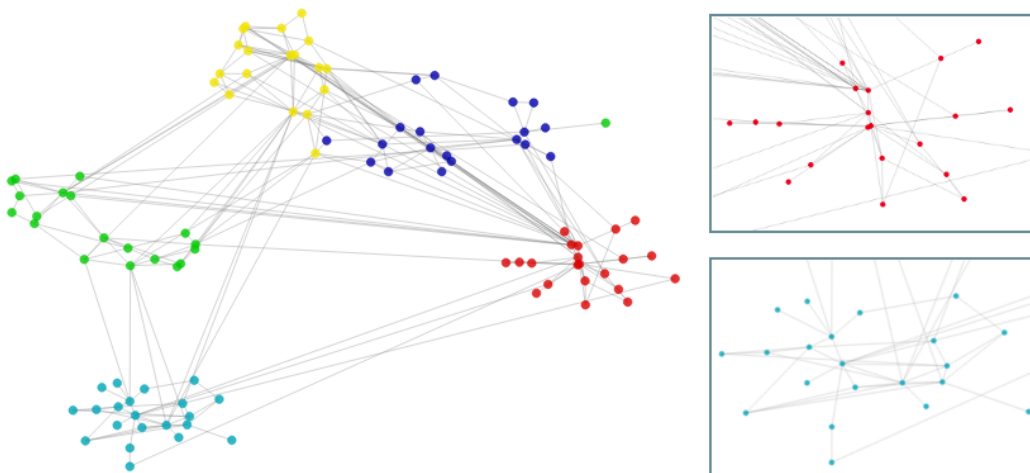

Figure 1: Trained PRODIGE model for a small subset of MNIST dataset, the full graph (left) and a zoom-in of two clusters (right). Vertex positions were computed using tSNE[30] over $d_G(\cdot, \cdot)$ distances; colors represent class labels. View interactively: `https://tinyurl.com/prodige-graph`

Figure 1 reveals several properties of the PRODIGE graphs. First, the number of edges per vertex is very uneven, with a large fraction of edges belonging to a few high degree vertices. We assume that these "popular" vertices play the role of "traffic hubs" for pathfinding in the graph. The shortest path between distant vertices is likely to begin by reaching the nearest "hub", then travel over the "long" edges to the hub that "contains" the target vertex, after which it would follow the short edges to reach the target itself.

Another important observation is that non-hub vertices tend to have only a few edges. We conjecture that this is the key to the memory-efficiency of our method. Effectively, the PRODIGE graph represents most of the data items by their relation to one or several "landmark" vertices, corresponding to graph hubs. Interestingly, this representation resembles the topology of human-made transport networks with few dedicated hubs and a large number of peripheral nodes. We plot the distribution of vertex degrees on Figure 3, which resembles power-law distribution, demonstrating "scale-free" property of PRODIGE graphs.

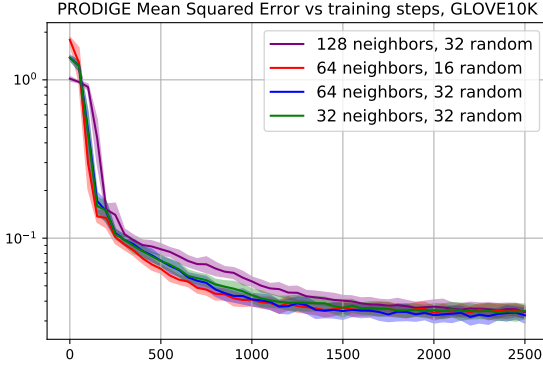

Figure 2: Learning curves, standard deviation over five runs shown in pale

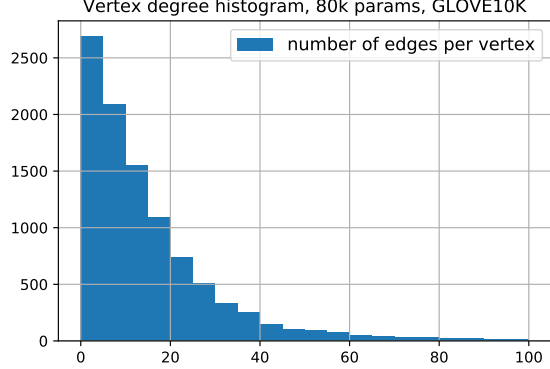

Figure 3: Vertex degree histogram for **GLOVE10K** with eight params/vertex

**Sanity check.** In this experiment we also verify that PRODIGE is able to reconstruct graphs from datapoints, which mutual distances are obtained as shortest paths in some graph. Namely, we generate connected Erdős–Rényi graphs with 10-25 vertices with edge probability $p=0.25$ and edge weights sampled from uniform $U(0, 1)$ distribution. We then train PRODIGE to reconstruct these graphs given pairwise distances between vertices. Out of 100 random graphs, in 91 cases PRODIGE was able to reconstruct all edges that affected shortest paths and all 100 runs the resulting graph had distances approximation error below $10^{-3}$.

## 4.2 Collaborative Filtering

In the next series of experiments, we investigate the usage of PRODIGE in the collaborative filtering task, which is a key component of modern recommendation systems.

Consider a sparse binary user-item matrix $F \in \{0, 1\}^{m \times n}$ that represents the preferences of $m$ users over $n$ items. $F_{ij}=1$ means that the $j$-th item is relevant for the $i$-th user. $F_{ij}=0$ means that the $j$-th item is not relevant for the $i$-th user or the relevance information is absent. The recommendation task is to extract the most relevant items for the particular user. A common performance measure for this task is HitRatio@k. This metric measures how frequently there is at least one relevant item among top-k recommendations suggested by the algorithm. Note that in these experiments we consider the simplest setting where user preferences are binary. All experiments are performed on the **Pinterest** dataset[31].

**Objective.** Intuitively, we want our model to rank the relevant items above the non-relevant ones. We achieve this via maximizing the following objective:

$$
\underset{G \sim p(G)}{E} L(G, \theta) = \underset{G \sim p(G)}{E} \quad \underset{u_i, x^+, x^-}{E} \log \frac{e^{-d_G(u_i, x^+)}}{e^{-d_G(u_i, x^+)} + \sum_{x^-} e^{-d_G(u_i, x^-)}} \tag{6}
$$

For each user $u_i$, this objective enforces the positive items $x^+$ to be closer in terms of $d_G(\cdot, \cdot)$ than the negative items $x^-$. We sample positive items $x^+$ uniformly from the training items that are relevant to $u_i$. In turn, $x^-$ are sampled uniformly from all items. In practice, we only need to run Dijkstra's algorithm once per each user $u_i$ to calculate the distances to all positive and negative items.

Similarly to the previous section, we speed up the training stage by considering only a small subset of edges. Namely, we build an initial graph using $F$ as adjacency matrix and add extra edges that connect the nearest users (user-user edges) and the nearest items (item-item edges) in terms of cosine distance between the corresponding rows/columns of $F$.

For this task, we compare the following methods:

- **PRODIGE-normal:** PRODIGE method as described above; we restrict a set of possible edges to include 16 user-user and item-item edges and all relevant user-item edges available in the training data;

- **PRODIGE-bipartite:** a version of PRODIGE that is allowed to learn only edges that connect users to items, counting approximately 30 edges per item. The user-user and the item-item edges are prohibited;
- **PRODIGE-random:** a version of our model that is initialized with completely random edges. All user-user, item-item, and user-item edges are sampled uniformly. In total, we sample 50 edges per each user and item;
- **SVD:** truncated Singular Value Decomposition of user-item matrix[4];
- **ALS:** Alternating Least Squares method for implicit feedback[32];
- **Metric Learning:** metric learning approach that learns the user/item embeddings in the Euclidean space and optimizes the same objective function (6) with Euclidean distance between user and item embeddings; all other training parameters are shared with PRODIGE.

The comparison results for two memory budgets are presented in Table 2. It illustrates that our PRODIGE model achieves better overall recommendation quality compared to embedding-based methods. Interestingly, the bipartite graph performs nearly as well as the task-specific heuristic with nearest edges. In contrast, starting from a random set of edges results in poor performance, which indicates the importance of initial edges choice.

| **Method** | PRODIGE | | | SVD | ALS | Metric Learning |
|---|---|---|---|---|---|---|
| | normal | bipartite | random | | | |
| $\leq$ **4 parameters per user/item** | | | | | | |
| HR@5 | **0.50213** | 0.48533 | 0.35587 | 0.33365 | 0.37005 | 0.45898 |
| HR@10 | **0.66192** | 0.64250 | 0.57492 | 0.49619 | 0.51815 | 0.60079 |
| $\leq$ **8 parameters per user/item** | | | | | | |
| HR@5 | **0.59921** | 0.57659 | 0.50113 | 0.5107 | 0.48617 | 0.54728 |
| HR@10 | **0.79021** | 0.77980 | 0.73485 | 0.70489 | 0.70075 | 0.74891 |

Table 2: HitRate@k for the Pinterest dataset for different methods and two memory budgets.

## 4.3 Sentiment classification

As another application, we consider a simple text classification problem: the algorithm takes a sequence of tokens (words) $(x_0, x_1, ..., x_T)$ as input and predicts a single class label $y$. This problem arises in a wide range of tasks, such as sentiment classification or spam detection

In this experiment, we explore the potential applications of learnable weighted graphs as an intermediate data representation within a multi-layer model. Our goal here is to learn graph edges end-to-end using the gradients from the subsequent layers. To make the whole pipeline fully differentiable, we design a projection of data items, represented as graph vertices, into feature vectors digestible by subsequent convolutional or dense layers.

Namely, a data item $v_i$ is represented as a vector of distances to $K$ predefined "anchor" vertices:

$$emb_G(v_i) = \langle d_G(v_i, v_0^a), d_G(v_i, v_1^a), \ldots, d_G(v_i, v_K^a) \rangle \tag{7}$$

In practice, we add the "anchor" vertices $\{v_0^a, \ldots, v_K^a\}$ to a graph before training and connect them to random vertices. Note that the "anchor" vertices do not correspond to any data object. The architecture used in this experiment is schematically presented on Figure 4.

Intuitively, the usage of PRODIGE in architecture from Figure 4 can be treated as a generalization of vectorial embedding layer. For instance, if a graph contains only the edges between vertices and anchors, this is equivalent to embedding $emb(v_i) = \langle w_\theta(e(v_i, v_0^a)), w_\theta(e(v_i, v_1^a)), \ldots, w_\theta(e(v_i, v_K^a)) \rangle$ with $O(n \cdot K)$ trainable parameters. However, in practice, our regularized PRODIGE model learns a more compact graph by using vertex-vertex edges to encode words via their relation to other words.

**Model and objective.** We train a simple sentiment classification model with four consecutive layers: embedding layer, one-dimensional convolutional layer with 32 output filters, followed by a global max pooling layer, a ReLU nonlinearity and a final dense layer that predicts class logits. Indeed, this model is smaller than the state-of-the-art models for sentiment classification and should be considered

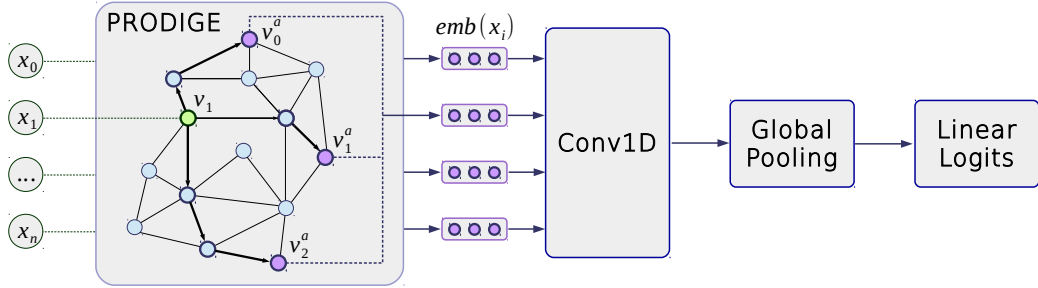

Figure 4: Model architecture for the sentiment classification problem. The PRODIGE graph is used as an alternative for the standard embedding layer, followed by a simple convolutional architecture.

only as a proof of concept. We minimize the cross-entropy objective by backpropagation, computing gradients w.r.t all trainable parameters including graph edges $\{\theta_w, \theta_b\}$.

**Experimental setup.** We evaluate our model on the IMDB benchmark [33], a popular dataset for text sentiment binary classification. The data is split into training and test sets, each containing $N=25,000$ text instances. For simplicity, we only consider $M=32,000$ most frequent tokens.

We compare our model with embedding-based baselines, which follow the same architecture from Figure 4, but with the standard embedding layer instead of PRODIGE. All embeddings are initialized with pre-trained word representations and then fine-tuned with the subsequent layers by backpropagation. As pretrained embeddings, we use GloVe vectors[5] trained on $25,000$ texts from the training set and select vectors corresponding to the $M$ most frequent tokens. In PRODIGE, the model graph is pre-trained by distance-preserving compression of the GloVe embeddings, as described in Section 4.1. In order to encode the "anchor" objects, we explicitly add $K$ synthetic objects to the data by running K-means clustering and compress the resulting $N + K$ objects by minimizing the objective (5).

| Representation | PRODIGE | GloVe | |
| --- | --- | --- | --- |
| | 100d | 100d | 18d |
| Accuracy | **0.8443** | **0.8483** | 0.8028 |
| Model size | **2.16 MB** | 12.24 MB | **2.20 MB** |

Table 3: Evaluation of graph-based and vectorial representations for the sentiment classification task.

For each model, we report test accuracy and the total model size. The results in Table 3 illustrate that PRODIGE learns a model with nearly the same quality as full 100-dimensional embeddings with much smaller size. On the other hand, PRODIGE significantly outperforms its vectorial counterpart of the same model size.

## 5   Discussion

In this work, we introduce PRODIGE, a new method constructing representations of finite datasets. The method represents the dataset as a weighted graph with a shortest distance path metric, which is able to capture geometry of any finite metric space. Due to minimal inductive bias, PRODIGE captures the essential information from data more effectively compared to embedding-based representation learning methods. The graphs are learned end-to-end via minimizing any differentiable objective function, defined by the target task. We empirically confirm the advantage of PRODIGE in several machine learning problems and publish its source code online.

**Acknowledgements:** Authors would like to thank David Talbot for his numerous suggestions on paper readability. We would also like to thank anonymous meta-reviewer for his insightful feedback. Vage Egiazarian was supported by the Russian Science Foundation under Grant 19-41-04109.

## Footnotes

[1] `https://github.com/stanis-morozov/prodige`

[2]We used a pre-trained gensim model "glove-wiki-gigaword-300"; see `https://radimrehurek.com/gensim/downloader.html` for details

[3]The dataset was obtained by running face_recognition package on CelebA images and uniformly sampling $10^4$ face embeddings, see `https://github.com/ageitgey/face_recognition`

[4]we use the Implicit package for both ALS and SVD baselines `https://github.com/benfred/implicit`

[5]We used `https://pypi.org/project/gensim/` to train GloVe model with recommended parameters

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
