[Reviews · NeurIPS 2019]

Reviewer 1



The paper proposes an embedding of data in a non-vectorial metric space, a weighted graph with shortest path distance. I think the paper is original. I find it well-written and the methodology, described in section 3, sound. The experimental results are not always impressive, though. The experiments in section 4.1 show that PRODIGE, the proposed technique, reaches good performance with low memory budget, but corresponding methods almost catch up with 8-dimensional embeddings only. Since nowadays we tend to work with much larger embeddings than 8-dimensional ones, I wonder in practice how important is this result. A similar reasoning applies to the experiments in section 4.3, in this case in addition I would suggest to compare PRODIGE with state-of-the-art embedding techniques such as BERT or ELMO. The experiments in section 4.2 looks more convincing. I appreciated the comparison with Metric Learning, the same technique using euclidean embeddings

Reviewer 2



[Concerns in Modeling] 1) Isomap was developed to extend the multidimensional scaling (MDS) by determining the neighbors of each point and measuring the distance by computing the distance by finding the shortest path via Dijkstra algorithm. Except the presence of each edge is probabilistic than deterministic, the core idea is quite similar to Isomap. The novelty should be better addressed by comparing to Isomap. 2) If there are many nodes and the graph is almost connected, is PRODIGE still scalable when finding the shortest paths between all pairs of nodes? 3) Typically, edges in the graph is far from being independent. For example, edges between words that frequently co-occur in the same contexts are not independent to each other. Edges between pixels in small coherent regions are not independent. Do we eventually need to know such dependency structures a priori to correctly represent arbitrary geometry in the data? 4) The suggested applications often require knowing the similarity in the original data space. However, in many cases, those similarity metrics are just the convenient choices rather than the correct metric. If we know the cosine similarity is the right metric to compare two different user-item preference vectors, for example, it automatically means our vanilla inner product space is the right realization of the data geometry. Can PRODIGE still learn the right/better geometry even if the original similarity metrics are incorrect? 5) The performance of Isomap is highly sensitive to the definition of neighbors (e.g., how many nearest elements will be declared as my neighbor). Is PRODIGE robust to such effect? As a related question, what is the reason to add synthetic anchor objects rather than using the known hub objects as anchors? 6) What is the purpose of adding random edges? Are 32-100 edges mentioned in Line 144 is general or specific to the particular tasks? 7) What happen if PRODIGE tries to represent already the data that is already represented by a graph? In other words, what happen if PRODIGE represent the graph data as its own graph with the same number of nodes? [Concerns in Evaluations] 1) I saw most of evaluations are done within highly limited memory budgets. Does PRODIGE show the same performance boost against the pure representations when we increase the budget to a usual laptop-sized memory? 2) What is the reason to tune the lambda parameter for matching the memory space rather than measuring the test error on the development set? 3) For the qualitative results by t-SNE visualization, is it true for all graph that non-hub vertices tend to have only a few edges? If hub vertices are defined as nodes with many edges, it is trivial that non-hub nodes have only a few edges. This property is generally called scale-free. It is better to quantitatively argue on the graph space by measuring whether the degree distribution follows the power-law. 4) Are SVD, ALS, and Metric Learning used in Table 2 the state-of-the-art approach for the collaborative filtering? If they are vanilla SVD, ALS, and metric learning, it may not be difficult to outperform. 5) What is the relation between the number of anchor objects and the performance in sentiment classification? Does increasing the dimension of the node embedding (by increasing K) keep improving the performance? ------------------------------------------------------- [After the author feedback] I thank the authors for their feedback to various concerns and additional experiments. I have read all other reviews and the corresponding feedback. Overall empirical results (especially the additional ones shown in the author feedback) seem useful and convincing, upgrading my original evaluation. But modeling the edge dependency and overall scalability from computing all-pair shortest paths is not clear. If the paper gets accepted, I encourage the authors to address these points in addition to adding many useful answers in the provided feedback.

Reviewer 3



Mainstream research in representation learning deals with representing structured data as vectorial data. In contrast, the authors propose to represent vectorial data as weighted graphs equipped with a shortest path distance. This makes the paper original. But, many graph construction methods have been proposed (e.g. \epsilon graphs, kNN graphs) given a similarity measure. Moreover, the similarity can be learned or graphs can be adapted to some task at hand. Here, the authors propose an end to end learning algorithm based on gradient descent which is also original. Experimental results are provided on tasks such as compression, classification and collaborative filtering. The development of the training objective is technically sound. But, in my opinion, many points are not carefully discussed. First, the probabilistic model assumes that edges are independent while a shortest path distance strongly depends on relations between edges. The training objective is an expectation over all random graphs and its estimation should be described in more details. It is not made clear in the paper that the results depend strongly on a good initialization, thus requiring a known problem-specific similarity. Last, convergence of the algorithm is not discussed. Therefore, the paper is based on an original idea and should lead to future publication. But, in my opinion, the paper is not ready for publication at NeurIPS because many questions on the learning algorithm remain unanswered. *** Detailed comments. * Introduction. It is not clear whether you discuss unsupervised or supervised representation methods ("for the task at hand", "in the specific task"). This remark also applies to the introduction of your paper. It is not clear whether you consider a target task. It is said in $3 of the related work section, a task specific loss is introduced in Section 3.2 and a problem-specific similarity is required in Section 3.3. * Related work. In my opinion, embeddings (whatever is the geometry) should be discussed in the introduction to clearly position your work. The related work section should be devoted to graph construction methods: base construction methods such as \epsilon graphs, kNN graphs, but also learning methods to adapt graphs to a target task. Also, metric learning should be discussed here. * $3.1. You assume independence between edges but you consider a shortest path distance in the learning objective. You should discuss this issue. * $3.1. You should state that there are 2n^2 parameters to be learned. * $3.2. You must estimate (3). This estimate seems to be based on only one sampling estimate. This is rather optimistic. Please elaborate on this and justify why such a rough estimate could be sufficient. Moreover, the interplay between gradient descent for weights and gradient descent for edge probabilities is not discussed. * $3.2. It is asserted that "once the training procedure converges, the output graph is nearly deterministic". Is this a theoretical statement or an experimental one ? Please, discuss the convergence of your algorithm. What can be said from a theoretical perspective ? Does-it converge in general ? What can be said on the output ? From an experimental perspective ? ... * $3.3. Here we learn that the algorithm is untractable. An heuristic is proposed. The heuristic is itself probabilistic therefore leading to questions on the convergence of your algorithm. Also here we learn that a problem-specific similarity is required for the method to work. * $4.1. The initialization heuristic of section 3.3 is for edge probabilities. Do you make several experiments with different draws ? Also what is the initialization procedure for edge weights ? Also we would like to know how you tune \lambda ? I.e. what is your crtiterion to choose \lambda ? What is the chosen value ? And how vary the output graphs depending on the initialization and on \lambda ? * $4.2. Metric learning is a research domain in itself. I do not understand what you called "metric learning" in this section. Which metric ? Which algorithm ? =============== after rebuttal =========== Thanks for your answers. I still have some concerns about independence of edge probabilities and the importance of the supposed given metric.

[Author Response · NeurIPS 2019]

We thank the reviewers for their constructive feedback and address their comments below.

**(R1, R2) Larger memory budgets.** In this paper, we focus on the models with low memory budgets. Such models are
needed for mobile devices, embedded systems, etc. However, as requested by the reviewers, we perform an additional
comparison with a budget of 32 parameters per datapoint. The results from Table 1 (bottom) demonstrate that the
advantage of PRODIGE over vectorial counterparts persists in this operating point as well.

**(R3) Why does PRODIGE become deterministic?** This property is induced by our sparsity regularizer (2) that
minimizes the $L_0$ norm of the adjacency matrix. This regularizer was originally proposed in [1], which explains why
the solutions are deterministic. Empirically, we also observe that edge probabilities converge to 0 or 1.

**(R2,R3) On independence of edges:** Yes, in our model edge indicators are independent random variables. However,
the training algorithm jointly optimizes all edges in the graph. Therefore, PRODIGE can learn arbitrary graph by setting
the probability of kept edges to 1 and the rest to 0. Thus, the independence assumption does not reduce the expressive
power of our model.

**(R3) Sampling efficiency.** The actual distance between two datapoints only depends on the existence of edges along
the shortest path and the absence of edges that would have led to an even shorter path if they were kept. There are
typically only a few such edges (e.g. about 15 for MNIST10K). We leverage this fact in lines 120-123 to propose an
objective (4) for more efficient optimization.

**(R2) PRODIGE vs Isomap.** Isomap constructs knn- or $\epsilon$-graph, then computes shortest paths in that graph and
approximates these paths with Multi-Dimensional Scaling in Euclidean space. Hence, Isomap produces vectorial
embeddings, while PRODIGE does not. Moreover, Isomap is specific to manifold learning and can only be applied to
task 4.1. We compare PRODIGE with Isomap (scikit-learn implementation), see Table 1 (top), which demonstrates the
advantage of our approach. Furthermore, PRODIGE is a general method that works for a variety of tasks (e.g. 4.2, 4.3).
If accepted, we will include a more detailed comparison of the two methods with explanation.

**(R2) SVD/ALS baselines in task 4.2** We also tried several neural collaborative filtering methods, but neither of them
was competitive with SVD and ALS. In fact, a recent study [2] demonstrates that these simple baselines perform
remarkably well for recommendation task on a variety of datasets, including Pinterest.

**(R3) The importance of initialization.** We investigate this issue in task 4.2, see lines 237-242 and Table 2 (original
paper). While the initialization is important, a completely random initialization still performs on par with most baselines.

**(R2,R3) Robustness and sensitivity to hyperparameters.** We verify the robustness of our training procedure by
running several experiments with different random initializations and different initial numbers of neighbors. Figure
1 shows the learning curves of PRODIGE under various conditions. While these results exhibit some noise from
Stochastic Gradient Descent, the overall training procedure is stable and robust.

| Method | MSE | Params |
|---|---|---|
| PRODIGE | **0.03356** | 76.5k |
| MDS | 0.05584 | 80k |
| Poincare MDS | 0.04839 | 80k |
| Isomap | 0.28242 | 80k |
| PRODIGE | **0.01031** | 292k |
| MDS | 0.01297 | 320k |
| Poincare-MDS | 0.01178 | 320k |

Table 1: Additional evaluations for task 4.1 performed on the **GLOVE10K** data

Figure 1: Learning curves, standard deviation over 10 runs shown in pale

Figure 2: Vertex degree histogram for **GLOVE10K** with 8 params/vertex

**(R2) Reconstructing known graphs with PRODIGE.** For this experiment, we generate connected Erdős–Rényi
graphs with 10-25 vertices with edge probability $p = 0.25$ and edge weights sampled from uniform $U(0,1)$ distribution.
We then train PRODIGE to reconstruct these graphs. Out of 100 random graphs, in 91 cases PRODIGE was able to
reconstruct all edges that affect shortest any paths and all 100 runs the resulting graph had MSE below $10^{-3}$. We agree
that this experiment is a useful sanity check and we will add it in the final version, if accepted.

**(R3) Metric learning details:** See lines 245-247. **(R2, R3) Tuning $\lambda$ :** We tune $\lambda$ to fit into the memory budget and
compare methods with the same budget. **(R2) Do we need to know dependency structures a priori?** No, PRODIGE
learns dependency structure from data. **(R2) Paths between all node pairs:** This can be performed efficiently with
an algorithm[3]. **(R2) Increasing K in 4.3:** We observe that for the same memory budget, larger K result in almost
equivalent performance. **(R2) Using known hubs as anchors in 4.3:** We tried that, however using K most frequent
tokens as anchors results in similar performance. We opted for synthetic anchors for generality; **(R2) Demonstrate**
**scale-free:** We observed the scale-free property, see Figure 2. Will add this in the final version, if accepted.

[1] Christos Louizos et al. Learning sparse neural networks through $L_0$ regularization. In *ICLR, 2018*.
[2] Ferrari Dacrema et al. Are we really making much progress? a worrying analysis of recent neural recommendation
approaches. In *RecSys, 2019*.
[3] Sebastian Knopp et al. Computing many-to-many shortest paths using highway hierarchies. In *ALENEX, 2007*.


[Meta-Review · NeurIPS 2019]

The paper proposed a quite interesting idea of representing data by weighted graphs (shortest path between nodes). Reviewers have raised concerns on edge dependency and the given similarity metric. However, I’m less worried about making the independence assumption because after all, it’s a model, and it seems to work well in experiments. Likewise, it is also common in variational inference to use independent distribution to approximate a graphical model, based on which learning is carried out. What interests me more is the general methodology of optimization. Suppose we want to optimize f(x). Instead of applying standard optimization, let us endow x_i with a Bernoulli distribution p_i, and then optimize E[f(x)] over all p_i by SGD/Adam/… The idea is surely straightforward, but my question is: has this been done *effectively* in existing machine learning practice? Example obstacles include high variance, computational cost, etc, which the paper has explicitly addressed by tapping into the sparsity of the shortest path solution. So I think this paper is a good addition to the conference. Technically, I wish the final version can clarify the following question regarding the p_\theta(b_i) term in Equation 2. First of all, it should be summation over all edges, rather than all instantiations of b_i because that will give a constant 1. Now if we write out the expectation and utilize the independence, each edge will contribute a term [p_\theta(0)]^2 + [p_\theta(1)]^2. I can't see how this term will encourage p_\theta(0) and p_\theta(1) to be either close to 0 or 1, because it is minimized at uniform distribution (modulo the parameterization \sigma(\theta_b)).